# The Impact of Interpreting Training Experience on the Attentional Networks and Their Dynamics

**DOI:** 10.3390/brainsci13091306

**Published:** 2023-09-11

**Authors:** Shunjie Xing, Jing Yang

**Affiliations:** 1Bilingual Cognition and Development Lab, Center for Linguistics and Applied Linguistics, Guangdong University of Foreign Studies, Guangzhou 510420, China; josephxing@foxmail.com; 2School of International Studies, Zhejiang University, Hangzhou 310058, China

**Keywords:** interpreting training, attention, alerting network, orienting network, executive network, working memory, bilingualism

## Abstract

Interpreting, a complicated and demanding bilingual task, depends heavily on attentional control. However, few studies have focused on the interpreters’ advantages in attention, and the findings so far have been inconsistent. Meanwhile, the connection between attentional networks and other cognitive abilities, such as working memory (WM), has rarely been explored in interpreters. The present study investigated whether interpreting experience (IE) contributed to the attentional networks of bilinguals and explored the link between interpreters’ attention and WM. Three groups of Chinese–English bilinguals, differing only in their duration of interpreting training (the More-IE group, the Less-IE group, and the No-IE group), completed the Attention Network Test (ANT). Results showed that only the alerting network was more efficient in the More-IE group than in the Less-IE and No-IE groups; moreover, the dynamics between the alerting and executive networks were significant only in the More-IE group. Furthermore, we found a negative correlation between the executive effect and the working memory capacity (WMC) in the More-IE group. Our study validated and provided empirical support for the Attentional Control Model, stimulating further research into neurocognitive mechanisms of advanced second language learning.

## 1. Introduction

Interpreting is a highly demanding bilingual task in which interpreters continuously switch between two activated languages and rapidly translate the source language into the target language under extreme time pressure. Therefore, working memory (WM), the ability to quickly retain and process verbal information, resolve interference, and coordinate multiple tasks within a short period, is essential for interpreters. Interpreting training may exercise WM over the long term and improve the storage and central executives of WM [1]. A large number of studies have shown that interpreters outperformed non-interpreters in working memory capacity (WMC) [2,3,4,5,6], which involves the ability to temporarily store and simultaneously process information, in WM span [7,8], which focuses solely on the ability to temporarily store information, as well as in central executives [9,10,11,12], while others failed to find the related evidence [13,14,15,16,17].

The discrepancy exists because WM may not be a fundamental ability that interpreting depends on. According to Dong and Li’s Attentional Control Model [18], the underlying mechanism for language control and processing control in interpreting is attention, which coordinates WM and language processing. Attention, one of the primary cognitive processes in humans, refers to the ability to selectively focus on the relevant information and ignore irrelevant information. It enables humans to allocate their limited cognitive resources to the most immediate tasks [19] and prioritize the computations of brain networks to achieve consciousness and observable behavior [20].

Attention and WM have increasingly been recognized as interconnected [21,22]. Specifically, attention is necessary for WM since it facilitates the activation, maintenance, and manipulation of the information stored in the mind [23,24]. Meanwhile, WM actively directs attention to relevant behavioral information [25]. Cowan [26,27] connected these two concepts in his Embedded-processes Model, defining WM as the capacity-limited focus of attention. Neurological evidence has also suggested shared brain mechanisms underlying the control of WM and attention in the prefrontal cortex and the bilateral insula [28,29].

Based on the Attentional Control Model [18], focused attention governs language control in interpreting tasks via establishing, reinforcing, and adjusting cognitive frameworks through WM and other cognitive functions. Simultaneously, divided attention facilitates processing control via coordination and WM, ensuring that interpreters sustain processing priorities and prevent interference from irrelevant information during interpreting tasks. Given that attention is the underlying mechanism for interpreting control, and that some studies revealed the influence of long-term bilingual language experience on attentional networks [30,31,32], interpreting training may tune interpreters’ attention, a topic that has received little attention to date.

As Petersen and Posner [33,34] define, the human attentional system consists of three independent, but unified, functional networks: the alerting network, the orienting network, and the executive network. The alerting network is responsible for attaining and maintaining the alert state [35]. It is essential for optimizing performance in tasks requiring higher cognitive functions [36]. The orienting network underlies the selection of stimuli from diverse sensory inputs. This selection can be exogenous when a specific target captures a person’s attention, or endogenous when a person voluntarily directs their attention to a particular target [37]. The executive network refers to interference control, such as monitoring and resolving interference. This network assists in identifying potential interference and acting appropriately based on the specific target configuration [38,39]. Although these three attentional subnetworks are functionally and anatomically independent [40,41], they can cooperate and work together [42,43], which is termed dynamics between networks [44,45]. The three networks are measured by the Attention Network Test (ANT) created by Fan et al. [46]. This task perfectly combines the cue-reaction time task [47] and the typical flanker task [48]. Briefly, participants indicated the direction of a central arrow flanked by arrows heading in the same direction or a different direction. Visual cues presented before the arrows guided the participants’ attention to where the arrows subsequently appeared.

So far, only five studies have been conducted on the attentional networks of interpreters using the ANT and the findings are inconsistent. Morales and her coworkers [44] measured the three attentional networks in professional interpreters and proficient bilinguals. Although this study failed to identify any group difference in mean reaction times (RTs), error rates, or any effect of attentional networks, it revealed group differences in the dynamics between the alerting and orienting effects. Specifically, the proficient bilinguals demonstrated a larger orienting effect in the presence of an alerting cue, whereas the orienting effect of professional interpreters was unaffected. This finding suggested that professional interpreters had a highly efficient focus of attention, ensuring that the state of alertness did not influence their orienting network. Woumans and her collaborators [49] focused on the orienting and executive effects among monolinguals, unbalanced and balanced bilinguals, and student interpreters. Regarding mean RTs and error rates, they reported non-significant differences between student interpreters and balanced bilinguals in the two attentional networks, consistent with the findings on professional interpreters and well-matched multilinguals by Babcock and Vallesi [50]. A two-year longitudinal study by the same team [7] traced three groups of participants: one group received interpreting training, and the other two groups received written translation or non-language training. The only relevant finding from this study showed faster executive network responses in all three bilingual groups after their respective training. Nevertheless, Babcock and her team members [7,50] conducted no additional interaction analysis to investigate the attention network dynamics in interpreters. Nour and her colleagues [45] also failed to reveal any effect of the attentional network between student interpreters and student translators before and after their master’s program. Their study, however, discovered the dynamics between alerting and executive effects. In particular, student interpreters exhibited a larger executive effect in the presence of an alerting cue as well as increased error rates compared to student translators. Interestingly, this study also included a group of professional interpreters in the post-training experiment, who demonstrated a larger executive effect and lower error rates when compared to student interpreters and translators.

The inconsistent findings regarding the interpreters’ advantage in attentional networks might be caused by insufficient sample size and unmatched bilingual background. First, small sample sizes may result in low statistical power, complicating the replication and reproduction of the same findings (28 student interpreters in Woumans et al. [49]; 16 professional interpreters in Morales et al. [44]; 23 professional interpreters in Babcock and Vallesi [50]; 10 student translators in Babcock et al. [7]; and 17 student interpreters in Nour et al. [45]). Even though the effect size is large, it may be inflated significantly [38,51]. Secondly, bilingual participants in those studies did not have equivalent levels of linguistic experience. Bilingualism is not a simple categorical label [52]; instead, it is a complex and dynamic process involving a range of experiences that contribute to distinct neurocognitive adaptations [53], such as L2 age of acquisition (AoA), language proficiency, and immersion experience. As Deluca and his collaborators [54] mentioned, together, those related factors modulate bilinguals’ cognitive and neural systems. In Woumans and her collaborators’ research [49], student interpreters exhibited a later AoA and a lower L2 proficiency than the balanced bilinguals. However, a later AoA and a lower L2 proficiency have been suggested to correlate with a less efficient executive network [55,56,57,58,59]. Further, a self-rated language proficiency questionnaire, rather than an objective and standardized measure, was used in all the above five studies for assessing L2 proficiency. Additionally, we observed that these five studies recruited participants with various L1 and L2 backgrounds, such as Dutch, French, Spanish, and German. Nevertheless, alphabetic languages were genetic relatives and none of these studies involved participants using two genetically unrelated languages, such as Chinese and English. In light of this, further studies with adequate sample sizes and background-matched participants may advance our understanding of attentional networks and their dynamics in interpreters.

To this end, the present study compared three groups of Chinese–English bilinguals, varying only in their interpreting experience (IE), labeled henceforward as the More-IE group, the Less-IE group, and the No-IE group. All participants were asked to complete the ANT created by Fan et al. [46]. Based on previous findings regarding interpreters and bilingual advantage in WM and attention, we expected that the More-IE group would outperform the other two groups in attentional networks and dynamics between those networks as a benefit of long-term interpreting training. We also hypothesized that WM and the attentional networks might be highly correlated in the More-IE group, as suggested in the Attentional Control Model [18], since the close link between WM and attention during interpreting tasks may be necessary for producing high-quality interpretations that accurately convey the intended meaning of the original discourse.

## 2. Materials and Methods

### 2.1. Participants

Ninety-six Chinese–English bilinguals (mean age = 23.23 ± 0.81 years, range = 22–25 years) participated in this study. They were all right-handed and had normal or corrected-to-normal vision and hearing abilities. The More-IE group (33 participants; 8 males) and the Less-IE group (33 participants; 3 males) were second-year postgraduate students majoring in English interpreting and translation, and the More-IE group completed a greater number of interpreting courses and after-class practice during their first year of postgraduate study. As a control group, the No-IE group (30 participants; 4 males) were second-year postgraduate students majoring in English literature. None of these participants had received formal training in interpreting or translation before their postgraduate studies. Data were collected when the participants began their second-year studies.

All participants self-reported their age, L2 AoA, L2 proficiency, L2 immersion, and L2 dominance in a brief version of the Language History Questionnaire (LHQ 3.0) [60]. Their English proficiency levels were measured by the Oxford Quick Placement Test [61]. Their WMCs were collected using the automated operation span task [62], a frequently used WMC test. As shown in Table 1, the three groups were matched in their L2 experiences, except for their interpreting training experiences. All participants provided informed consent and received monetary compensation upon completing the experiment. This study was approved by the Research Ethics Committee of the Bilingual Cognition and Development Lab at the Guangdong University of Foreign Studies, China.

### 2.2. Task and Procedure

We administered the ANT developed by Fan et al. [46] to assess the participants’ alerting, orienting, and executive networks. As illustrated in Figure 1, each trial started with a fixation cross (+) lasting between 400 ms and 1600 ms. Then, an attentional cue was presented for 100 ms, allowing us to assess the alerting and orienting networks. Those cue types included the center cue (an asterisk at the central location of the fixation cross), the double cue (an asterisk above and below the central fixation cross), and the spatial cue (an asterisk presented above or below the fixation cross) conditions. Otherwise, the attentional cue may not appear at all, known as the no-cue condition.

Following a subsequent short fixation period of 400 ms, participants were presented with five horizontal arrows, either above or below the fixation cross, with arrowheads pointing to the left (←) or right (→). They were asked to press the left or right button as accurately and rapidly as possible to indicate the direction of the center arrow, which is flanked by two additional arrows pointing in the same direction (i.e., congruent condition), in the opposite direction (i.e., incongruent condition), or by single lines (i.e., neutral condition). These arrows remained on the screen until participants made a response. After the response time limit of 1700 ms, a post-target fixation was presented for a variable duration based on the first fixation’s duration and RT (3500 ms–RT–the first fixation time). The subsequent trial began after this interval. The ANT comprised a 24-trial practice block with feedback and three 96-trial experimental blocks without feedback. Each experimental block consisted of two sets of 48 trials (4 attentional cues × 3 flanker types × 2 target locations × 2 target directions). The fixation cross remained at the center of the screen throughout the experiment, and the presentation order of the trials within each block was randomized.

### 2.3. Data Collection and Analysis

The ANT was programmed using E-Prime 3.0 (Psychology Software Tools Inc., Pittsburgh, PA, USA), and data were collected individually. The mean RTs of correct trials and error rates (mean percentage of error trials) were further analyzed. Outliers above or below 2.5 standard deviations from the mean RTs were excluded from the analysis.

#### 2.3.1. Analyses of Attentional Networks and Their Effects

The general analysis on the ANT included two within-subject variables (attentional cue: no cue, double cue, center cue, and spatial cue; flanker type: congruent condition, incongruent condition, and neutral condition) and one between-subject variable (group: More-IE group, Less-IE group, and No-IE group). Following previous practices in studies using the ANT, we focused on separate two-way mixed-design ANOVAs relevant to assessing the three networks. For the alerting network analysis, two-way mixed-design ANOVAs were conducted with one between-subject variable (group: More-IE group, Less-IE group, and No-IE group) and one within-subject variable (attentional cue: no cue vs. double cue). For the orienting network analysis, similar two-way mixed-design ANOVAs were conducted with one between-subject variable (group: More-IE group, Less-IE group, and No-IE group) and one within-subject variable (attentional cue: center cue vs. spatial cue). For the executive network analysis, two-way mixed-design ANOVAs involved one between-subject variable (group: More-IE group, Less-IE group, and No-IE group) and one within-subject variable (flanker type: congruent condition vs. incongruent condition).

After each network analysis, we conducted multiple one-way ANOVAs to compare the effect of each attentional network among the three groups. It is measured as a comparison of the performance between one condition and the corresponding reference condition, according to the formulas below [46]:Alerting effect = (RTs/Error rates _no cue_ – RTs/Error rates _double cue_)Orienting effect = (RTs/Error rates _center cue_ – RTs/Error rates _spatial cue_)Executive effect = (RTs/Error rates _incongruent condition_ – RTs/Error rates _congruent condition_)

Larger alerting and orienting effects indicated better performances facilitated by an attentional cue. In contrast, a larger executive effect indicated an inferior performance of the executive network.

#### 2.3.2. Dynamics Analyses between the Attentional Networks

We performed two separate three-way mixed-design ANOVAs for mean RTs and error rates to explore the dynamics between attentional networks associated with interpreting training experience. Like the attentional network analyses conducted in Section 2.3.1, the dynamics analyses were also integral parts of the general analysis chosen for our research interest, as done by Nour et al. [45] and Morales et al. [44].

The first analysis focused on the Interaction between the alerting and executive networks with one between-subject variable (group: More-IE group, Less-IE group, and No-IE group) and two within-subject variables (attentional cue: no cue vs. double cue; flanker type: congruent condition vs. incongruent condition).

The second analysis on the interaction between the orienting and executive networks entailed one between-subject variable (group: More-IE group, Less-IE group, and No-IE group) and two within-subject variables (attentional cue: center cue vs. spatial cue; flanker type: congruent condition vs. incongruent condition).

Since both the alerting and orienting networks were monitored by attentional cues, the dynamics between the two attentional networks could not be examined in the current ANT version [46].

#### 2.3.3. Correlation Analyses

In line with prior studies on the ANT, we conducted bivariate Pearson correlation analyses to examine the functional independence between the three networks within each group. These correlation analyses allowed us to assess whether the alerting, orienting, and executive effects were independent of each other within each group.

Additionally, if we found that the three attentional networks were indeed independent within each group, we planned to conduct additional bivariate Pearson correlation analyses to explore the relationship between the attentional networks and WM within each group.

Moreover, in the event of finding a significant correlation between any attentional effect and WMC within any group, we intended to perform subsequent regression analyses to further examine if WMC significantly predicted performance in the relevant attentional network within that specific group.

## 3. Results

### 3.1. Attentional Networks and Their Effects

The analyses of attentional networks are summarized in Table 2.

#### 3.1.1. Alerting Network

The alerting network analysis of RT data revealed the main effect of the attentional cue, *p* < 0.001, *η*_p_^2^ = 0.83, indicating faster responses for the trials preceded by the double cue than for trials preceded by no cue. There was no significant group effect. The interaction between the attentional cue and group was statistically significant, *p* = 0.037, *η*_p_^2^ = 0.07. Simple effect analysis showed that the More-IE group responded faster in the presence of the double cue compared to the presence of no cue (*t* = 3.43, *p* = 0.001, Cohen’s *d* = 0.84), as did the Less-IE group (*t* = 2.88, *p* = 0.005, Cohen’s *d* = 0.71) and the No-IE group (*t* = 2.39, *p* = 0.02, Cohen’s *d* = 0.62). 

We observed no significant effect in the analysis of the alerting network using error rates (*F*s < 1, *p*s > 0.5).

Notably, as indicated by the significant interaction, the one-way ANOVA on the alerting effect showed that the three groups performed differently in mean RTs (see Figure 2a), *F*(2, 95) = 3.41, *p =* 0.037. Post hoc analysis further indicated that the More-IE group benefited more from the presence of an alerting cue than the Less-IE group (*p* = 0.027) and the No-IE group (*p* = 0.026), while no difference in the alerting effect was found between the Less-IE group and the No-IE group (*p* = 0.935). We did not observe a significant main effect when analyzing error rates, *F*(2, 95) = 1.74, *p =* 0.181, suggesting that the presence of an alerting cue did not influence the correctness of the participants’ responses (see Figure 2b).

#### 3.1.2. Orienting Network

Our analysis of the orienting network revealed the main effect of attentional cue, *p* < 0.001, *η*_p_^2^ = 0.83, suggesting that the presence of the spatial cue elicited faster responses compared to the presence of the center cue (the More-IE group, *t* = 2.60, *p* = 0.012, Cohen’s *d* = 0.64; the Less-IE group, *t* = 2.72, *p* = 0.008, Cohen’s *d* = 0.67; and the No-IE group, *t* = 2.63, *p* = 0.011, Cohen’s *d* = 0.68). The group main effect was not significant, nor was the interaction between attentional cue and group.

In terms of error-rate analysis, we observed the main effect of attentional cue, *p* < 0.001, *η*_p_^2^ = 0.15, indicating more accurate responses to the spatial-cue trials relative to the center-cue trials (the More-IE group, *t* = 1.6, *p* = 0.114, Cohen’s *d* = 0.4; the Less-IE group, *t* = 1.55, *p* = 0.127, Cohen’s *d* = 0.38; and the No-IE group, *t* = 1.89, *p* = 0.064, Cohen’s *d* = 0.49). Neither the main effect of the group nor the interaction between attentional cue and the group reached significance (*Fs* < 1, *ps* > 0.5).

The analyses conducted on the orienting effect revealed no significant difference among the three groups, both in terms of the mean RTs and error rates (*Fs* < 1, *ps* > 0.8), suggesting that the benefits induced by the orienting cue were similar for the three groups (see Figure 2).

#### 3.1.3. Executive Network

Our analysis of the executive network identified the main effect of the flanker type, *p* < 0.001, *η*_p_^2^ = 0.93, indicating slower responses to the incongruent trials than congruent trials (the More-IE group, *t* = 5.85, *p* < 0.001, Cohen’s *d* = 1.44; the Less-IE group, *t* = 6.09, *p* < 0.001, Cohen’s *d* = 1.5; and the No-IE group, *t* = 5.46, *p* < 0.001, Cohen’s *d* = 1.411). However, neither the group’s main effect, nor the interaction between the flanker type and group were significant.

We conducted the same ANOVA with the error rates and found the significant main effect of flanker type, *p* < 0.001, *η*_p_^2^ = 0.44, indicating more incorrect responses to the incongruent trials relative to congruent trials (the More-IE group, *t* = 5.39, *p* < 0.001, Cohen’s *d* = 0.83; the Less-IE group, *t* = 4.88, *p* < 0.001, Cohen’s *d* = 0.76; and the No-IE group, *t* = 3.72, *p* < 0.001, Cohen’s *d* = 0.62). Neither the main effect of the group nor the interaction between the attentional cue and the group reached significance (*Fs* < 1.5, *ps* > 0.2).

The analyses of the executive effect indicated no significant difference among the three groups, as evidenced by the mean RTs and error rates (*Fs* < 1.5, *ps* > 0.2). Thus, the interference from the incongruent flankers was comparable across the three groups of participants (see Figure 2).

### 3.2. Dynamics between the Attentional Networks

We conducted a three-way mixed-design ANOVA involving the group (More-IE group, Less-IE group, and No-IE group), attentional cue (no cue vs. double cue), and flanker type (congruent condition vs. incongruent condition) factors to examine the interaction between alerting and executive networks. The main effects of attentional cue, *F*(1, 93) = 209.62, *p* < 0.001, *η*_p_^2^ = 0.69, and flanker type, *F*(1, 93) = 910.85, *p* < 0.001, *η*_p_^2^ = 0.91, were both significant, suggesting faster responses in the double-cue trials and the congruent trials. The interaction between attentional cue and flanker type also reached significance, *F*(1, 93) = 29.89, *p* < 0.001, *η*_p_^2^ = 0.24, revealing a larger executive effect in double-cue trials compared to no-cue trials (*t* = 4.75, *p* < 0.001, Cohen’s *d* = 0.69). We also found a significant interaction between the flanker type and the group, *F*(2, 93) = 3.24, *p* = 0.044, *η*_p_^2^ = 0.07, suggesting a larger executive effect for the More-IE group when compared to the Less-IE group (*p* = 0.021) and the No-IE group (*p* = 0.022). Additional RT analysis of the executive effect revealed that the three groups performed differently only under the double-cue condition, *F*(2, 95) = 3.34, *p* = 0.04. Specifically, the More-IE group exhibited a larger executive effect in the presence of an alerting cue than the Less-IE group (*p* = 0.029) and the No-IE group (*p* = 0.027); no significant difference was found between the Less-IE group and the No-IE group (*p* = 0.934) (see Figure 3a). Neither the main effect of the group nor the other interactions reached significance (all *ps* > 0.1).

The error analysis of the interaction between alerting and executive networks revealed the main effect of the flanker type, *F*(1, 93) = 65.23, *p* < 0.001, *η*_p_^2^ = 0.41, suggesting that more errors were made during incongruent conditions than congruent conditions. The interaction between the attentional cue and flanker type was also significant, *F*(1, 93) = 7.06, *p* = 0.009, *η*_p_^2^ = 0.07, indicating a larger executive effect in the presence of an alerting cue compared to the no-cue condition (*t* = 2.11, *p* = 0.036, Cohen’s *d* = 0.31). We also conducted analysis in comparing the executive effect among the three groups under the two conditions. However, no significant main effect was observed (*Fs* < 2, *ps* > 0.1), suggesting that the three groups’ executive effect was similar under the two conditions when measured by error rates (see Figure 3b). There were no other statistically significant main effects or interactions (*ps* > 0.07).

Regarding the interaction between orienting and executive networks, we conducted a three-way mixed-design ANOVA of the group (More-IE group, Less-IE group, and No-IE group), attentional cue (center cue vs. spatial cue), and flanker type (congruent condition vs. incongruent condition) factors. The main effects of the attentional cue, *F*(1, 93) = 355.63, *p* < 0.001, *η*_p_^2^ = 0.79, and flanker type, *F*(1, 93) = 942.91, *p* < 0.001, *η*_p_^2^ = 0.91, were significant, indicating faster responses in the spatial-cue trials and the congruent trials. The interaction between the attentional cue and flanker type reached significance, *F*(1, 93) = 20.66, *p* < 0.001, *η*_p_^2^ = 0.18. A simple effect analysis revealed a larger executive effect in center-cue trials compared to spatial-cue trials (*t* = 3.86, *p* < 0.001, Cohen’s *d* = 0.56). Neither the main effect of the group nor the other interactions were significant (all *ps* > 0.2).

The error analysis of the interaction between the orienting and executive networks also showed the significant main effect of the attentional cue, *F*(1, 93) = 19.67, *p* < 0.001, *η*_p_^2^ = 0.18, and flanker type, *F*(1, 93) = 51.82, *p* < 0.001, *η*_p_^2^ = 0.36, suggesting that more incorrect responses were made under center-cue conditions and incongruent conditions. The interaction between the attentional cue and flanker type was also significant, *F*(1, 93) = 21.57, *p* < 0.001, *η*_p_^2^ = 0.19. A simple effect analysis indicated a larger executive effect without the presence of the orienting cue (*t* = 3.55, *p* < 0.001, Cohen’s *d* = 0.51). No other main effects or interactions were statistically significant (all *ps* > 0.4).

### 3.3. Correlations between Attention Networks and WM

Results showed no significant correlations between the alerting, orienting, and executive effects in the More-IE group (alerting effect × orienting effect: *r* = 0.069, *p* = 0.701; alerting effect × executive effect: *r* = −0.078, *p* = 0.665; and orienting effect × executive effect: *r* = −0.087, *p* = 0.632), in the Less-IE group (alerting effect × orienting effect: *r* = 0.013, *p* = 0.944; alerting effect × executive effect: *r* = −0.098, *p* = 0.589; and orienting effect × executive effect: *r* = 0.153, *p* = 0.396), and in the No-IE group (alerting effect × orienting effect: *r* = 0.086, *p* = 0.653; alerting effect × executive effect: *r* = 0.052, *p* = 0.786; and orienting effect × executive effect: *r* = 0.226, *p* = 0.229).

Regarding the relationship between WM and the three attentional networks, we found a significant negative correlation between WMC and executive effect in the More-IE group (*r* = −0.514, *p* = 0.002), but not in the Less-IE group (*r* = −0.062, *p* = 0.731) or the No-IE group (*r* = −0.178, *p* = 0.347). However, no significant correlation was found between alerting effect and WMC in any of the three groups (the More-IE group: *r* = −0.088, *p* = 0.625; the Less-IE group: *r* = 0.245, *p* = 0.17; and the No-IE group: *r* = −0.228, *p* = 0.225). Similarly, no significant correlation emerged between the orienting effect and WMC in the three groups (the More-IE group: *r* = 0.079, *p* = 0.662; the Less-IE group: *r* = −0.019, *p* = 0.916; and the No-IE group: *r* = −0.089, *p* = 0.641). Given the strong negative correlation between WMC and executive effect in the More-IE group, we subsequently conducted a regression analysis specifically within the More-IE group to explore whether WMC significantly predicted their performance of the executive network. The further linear regression analysis showed that WMC significantly predicted the performance of the executive network only for the More-IE group, *R^2^* = 0.24, *F*(1, 31) = 11.13, *p* = 0.002.

## 4. Discussion

The present study investigated the effects of interpreting training on bilinguals’ attentional networks and the relationship between interpreters’ attention and WM. Our results showed that only the alerting network was significantly more efficient in the More-IE group than in the Less-IE and No-IE groups; furthermore, the dynamics between the alerting and executive networks were significant only in the More-IE group. Additionally, we found a negative correlation between the executive effect and WMC in the More-IE group.

### 4.1. Interpreting Training Experience and Attentional Networks

Regarding the alerting network, this is the first study to identify interpreters’ advantages in achieving and maintaining an alerting state. This result echoed the earlier findings of Nour et al. [45], who reported a larger alerting effect in student interpreters than student translators (as a bilingual control group). However, the difference in the alerting network was not statistically significant in their study, possibly due to a small sample size and insufficient training experience. Interestingly, the More-IE group in the current study demonstrated a larger alerting effect without sacrificing accuracy. This finding corroborates the nature of interpreting tasks, in which bilinguals with intense interpreting training are expected to increase their sensitivity and readiness for numerous challenges while simultaneously ensuring the precise translation of languages at a highly alert state [18].

Moreover, the observed interpreter advantages in the alerting network may also be attributed to the requirements of the interpreting task. A few studies [63,64] have revealed a more efficient alerting network in bilinguals than in monolinguals. One possible explanation is that the alerting effect appears robust with language switching [65]. When it comes to interpreting tasks, interpreters must frequently switch between listening to the source language and speaking in the target language while searching for equivalent words, phrases, and clauses [66]. According to the adaptive control hypothesis by Green and Abutalebi [67], bilinguals’ adaptation of cognitive system is based on the interactional language context in which a bilingual is immersed. Interpreting features dense language switching under extreme time pressure, which in turn, necessitates a rapid adaptation of the cognitive system relative to other bilinguals. Consequently, the interpreters’ advantages in the alerting network may stem from the frequent and consistent switch between languages compared to general bilinguals.

Concerning the orienting network, the three groups showed similar orienting effects, suggesting no interpreter advantage over other general bilinguals. This finding is consistent with the results of the five ANT studies mentioned earlier. Nevertheless, it is possible that interpreters may have superior orienting ability since they must actively divert their attention to different verbal and non-verbal messages via auditory and visual modalities [27,68], then select them in terms of demands. Evidence for this claim derived from the interpreters’ advantages in cognitive flexibility [12,69] and coordination [70,71]. In the current study, two modes of orienting (i.e., the exogenous and endogenous control) were conflated in measuring the orienting network, which may obscure the potential advantages of interpreters. Even though the informative peripheral cues were used in ANT to capture attention reflexively (exogenous orienting), the information value would inevitably incur endogenous orienting [72]. Future research may apply the Combined Attention Systems Test, designed by Lawrence and his coworkers [73], to compute distinct orienting effects and probe more deeply into interpreters’ orienting networks.

Regarding the executive network, Arora and Klein [30] conducted a meta-analysis and reported a credible bilingual advantage in executive effect. A convincing explanation is that bilinguals must continuously resolve the interferences of their unintended languages when they try to comprehend and produce in another language [74]. Interpreting, as an extreme form of bilingual experience, may place demands on the executive network to a higher degree. Unlike general bilinguals, the active involvement of the source and the target language while interpreting leads to more intense cross-linguistic interference [39], interference at different levels of language processing [75], and even non-verbal interference, such as vocally produced noise and voices from different individuals [38].

Admittedly, our results cannot confirm the existence of a bilingual advantage in the executive effect due to the absence of a monolingual group in our study. However, we found no evidence for interpreter advantages in the executive network when compared to other bilinguals, which is in accordance with previous studies [14,17]. This result was unsurprising, given that the interference control advantage may rapidly diminish before participants respond in behavioral studies. Using the electroencephalography (EEG) recording, Dong and Zhong [11] recruited a More-IE group (with 2.3 years of interpreting training) and a Less-IE group (with 0.5 years of interpreting training), reporting interpreter advantages in interference control ability (as indexed by larger P3 amplitude). In addition, alpha oscillations were commonly associated with the executive network [76]. Focusing on this index, Yagura and his colleagues [77] found that the professional interpreters (with 15 years of interpreting training) exhibited more efficient executive networks than student interpreters (with 1 year of interpreting training). Therefore, a more sensitive tool, such as the EEG recording, is necessary to explore the developmental trajectory of the executive network in future research.

### 4.2. Dynamics between the Alerting and Executive Networks Related to Interpreting Training

The lack of correlation among the three networks demonstrated the independence of the attentional subsystems. However, our findings showed that the interpreting training experience correlates with the dynamics between the alerting and executive networks. This is in line with Nour et al. [45], suggesting an enhancement of interpreters’ executive effect in the presence of an alerting cue.

One possible explanation is that a larger alertness of the interpreters gained from the alerting cue facilitates information processing and elicits fast responses to input sensory modality, which, however, caused a larger executive effect [78,79]. Notably, the input sensory modality is a prerequisite for simultaneous interpreting and consecutive interpreting [8]. It is the core of language-modality connections, responsible for receiving verbal information and guaranteeing the flow of information from auditory to vocal [18]. Consequently, we may speculate that experienced interpreters can flexibly allocate their attentional resources under a highly alert state and prioritize the rapid and precise output of information from the input sensory modality, rather than concentrate on interference control [80].

It is also possible that the enhanced alertness of interpreters, especially when triggered by alerting cues, facilitates the global processing of stimuli, leading to less localized target processing [81,82]. ANT interference primarily occurs locally [45,83]. Therefore, the insufficient local interference processing resulted in a larger executive effect of the More-IE group, which may be more pronounced than the other two groups when the alerting cue was presented.

### 4.3. Interconnected Relationship between the Executive Network and WM in Interpreters

The present study found a negative correlation between WMC and executive effect in the More-IE group. Therefore, interpreters’ interference control ability was associated with a higher WMC, and vice versa. Our finding provides direct evidence for the interconnected nature of WM and attention as depicted in the Attentional Control Model [18].

We traced this finding back to the mechanism of interference control in interpreting. Bilinguals continuously experience interference from a non-target language; the issue of interference control in bilinguals centers on the specific control process to regulate their language use. The inhibitory view of language selection has gained much support. According to this perspective, the appropriate selection of languages requires continuous suppression of irrelevant languages [84,85]. However, this inhibition account may not apply equally to interpreting, especially in simultaneous interpreting, as interpreters frequently and precisely switch between the source language and the target language under challenging time constraints. Thus, the dominant view of inhibition is neither economical nor practical to meet the special requirement of interpreting [39].

An alternative mechanism of interference control optimal for interpreting is target enhancement, which refers to enhancing target activation to a higher degree [67,86] instead of inhibiting task-irrelevant information. In light of the target enhancement account, interpreters can better memorize and manipulate the representations of the target information as WMC increases. These advantages, brought about by the increasing WMC, should ensure greater task enhancement of what is required at the moment and achieve more effective interference control in interpreting. Our findings on the negative correlation between WMC and the executive effect of interpreters provided supporting evidence for this claim, suggesting that interference control underlying the executive network is mainly achieved through target enhancement in interpreting tasks and supported by WM, by which focused attention maintains and achieves language control [18].

### 4.4. Limitations and Future Study

Admittedly, the present study has several limitations to be considered and addressed in future research. Firstly, the causal relationship between interpreting training experience and the advantage in attentional networks merits additional longitudinal research to determine whether the advantage in attentional networks is a prerequisite or a consequence of the intense bilingual experience. Secondly, the group classification in our study is still insufficient to determine how slight changes in interpreting training influence performance. Therefore, further studies should examine interpreting training as a continuum. Variables that situate individuals along that continuum can be modeled to better understand how the dynamic nature of in-class and after-class interpreting training experience affects the attentional networks differently. Thirdly, most tasks used to measure individual differences in attention need better reliability [87,88,89]. Further study could utilize the three squared tasks developed by Burgoyne and his team members [90] to resolve critical theoretical and methodological flaws in tasks measuring attention, and these tasks can be administered in less than 3 min (compared to over 15 min for ANT). Fourthly, whether the relatedness between L1 and L2 influences interpreters’ advantages in attentional networks is unclear. Future studies could directly compare interpreters with different language backgrounds, such as Chinese–English and Dutch–French, to explore whether the advantages remain robust.

Accordingly, future research could trace the learning process of student interpreters over time and collect data at regular intervals to reveal the neurocognitive mechanism underlying advantageous attentional networks induced by interpreting training.

## 5. Conclusions

The present study investigated how interpreting training impacted attentional networks and explored the link between interpreters’ attention and WM. Results showed that the More-IE group outperformed the other two groups in the alerting network. Moreover, the dynamics between the alerting and executive networks were demonstrated only in the More-IE group. Specifically, the More-IE group exhibited a larger executive effect in the presence of an alerting cue. Additionally, we found a negative correlation between the executive effect and WMC associated with interpreting training experience. Our study provided empirical evidence for the Attentional Control Model [18]. Future longitudinal studies regarding training as a continuum will shed further light on the interpreters’ cognitive advantages and provide pedagogical implications for bilingual education.

## Figures and Tables

**Figure 1 brainsci-13-01306-f001:**
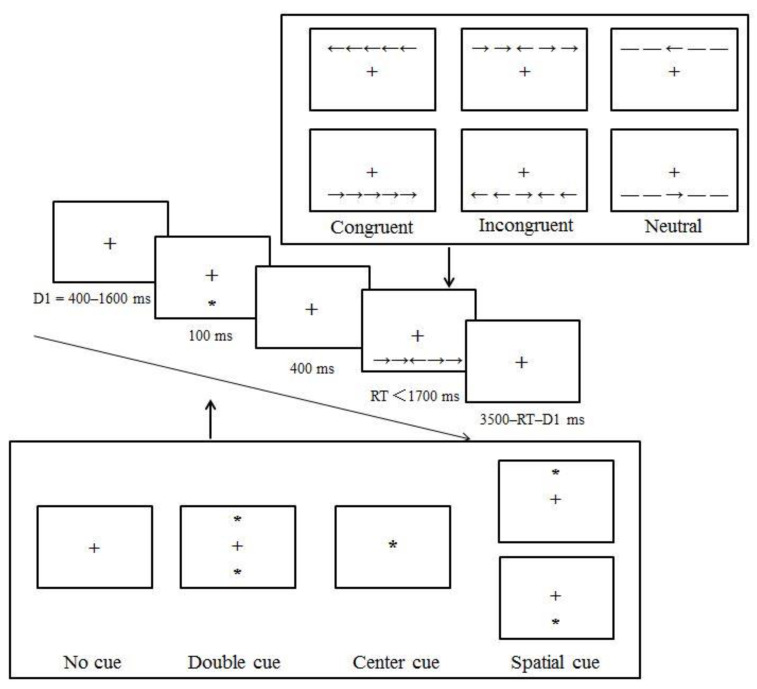
The Attention Network Test (ANT) administered in the current study. D1 refers to the fixation period for the random variable duration (400–1600 ms).

**Figure 2 brainsci-13-01306-f002:**
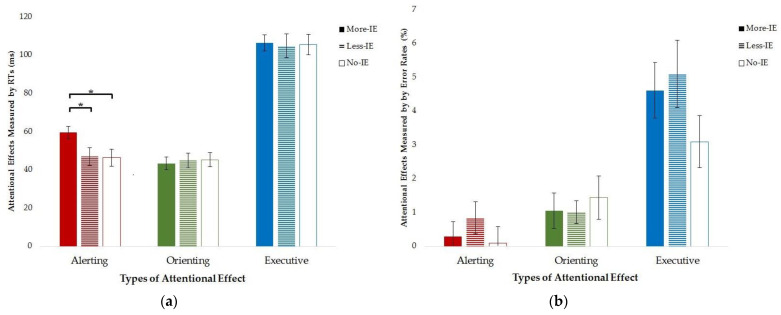
The effects of attentional networks of the More-IE, Less-IE, and No-IE groups in terms of the mean RTs (**a**) and error rates (**b**). The columns with red, green, and blue colors indicate different attentional effects (Red columns, alerting effect; Green columns, orienting effect; and Blue columns, executive effect). *, *p* < 0.05.

**Figure 3 brainsci-13-01306-f003:**
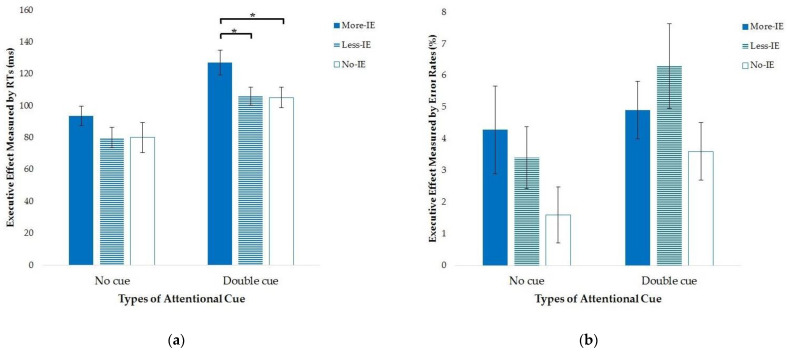
The executive effect of the More-IE group, Less-IE group, and No-IE groups in the presence of the no cue and the double cue measured by the mean RTs (**a**) and error rates (**b**). *, *p* < 0.05.

**Table 1 brainsci-13-01306-t001:** Participants’ background information (standard deviation).

	More-IE Group (N = 33)	Less-IE Group (N = 33)	No-IE Group (N = 30)
Age (years)	23.30 (0.73)	23.27 (0.91)	23.10 (0.8)
L2 AoA	9.09 (2.71)	8.52 (3.10)	8.70 (2.65)
L2 immersion (range: 0–1)	0.62 (0.12)	0.61 (0.17)	0.67 (0.32)
L2 dominance (range: 0–1)	0.39 (0.08)	0.38 (0.11)	0.38 (0.08)
Self-rated L1 proficiency (range: 0–1)	0.79 (0.13)	0.80 (0.14)	0.80 (0.15)
Self-rated L2 proficiency (range: 0–1)	0.68 (0.12)	0.63 (0.13)	0.65 (0.12)
L2 proficiency level (range: 0–60)	43.61 (9.70)	43.39 (8.55)	43.33 (7.61)
WMC (range: 0–75)	53.09 (15.4)	45.91 (12.68)	49.24 (15.22)
In-class interpreting training (hours) ***	263.27 (14.47)	47.71 (10.14)	-
After-class interpreting practice (hours) ***	181.33 (139.58)	107.54 (144.91)	-

Note: L1 = first language; L2 = second language; and AoA = age of acquisition. Age, L2 AoA, L2 immersion, L2 dominance, self-rated L1 proficiency, and self-rated L2 proficiency were obtained via the Language History Questionnaire (LHQ 3.0) [60]. All the indexes in the LHQ 3.0 are continuous variables ranging from 0 to 1 based on the standard algorithm. L2 proficiency level was measured by the Oxford Quick Placement Test [61]. WMC was assessed by the automated operation span task [62]. Interpreting training experiences included in-class and after-school practices, estimated based on first-year class and self-report after-class practice hours. *** *p* < 0.001.

**Table 2 brainsci-13-01306-t002:** Summary of attentional network analyses.

		Main Effect of Cue/Flanker			Main Effect of Group			Interaction Effect		
		*F*(1, 93)	*p*	*η* _p_ ^2^	*F*(2, 93)	*p*	*η* _p_ ^2^	*F*(2, 93)	*p*	*η* _p_ ^2^
Alerting Network	RTs (ms)	466.61	<0.001	0.83	2.11	0.127	0.04	3.41	0.037	0.07
	Error Rates	0.36	0.552	0.004	0.67	0.517	0.01	0.48	0.619	0.01
Orienting Network	RTs (ms)	463.75	<0.001	0.83	1.52	0.224	0.03	0.08	0.919	0.002
	Error Rates	15.87	<0.001	0.15	0.53	0.59	0.01	0.21	0.813	0.004
Executive Network	RTs (ms)	1173.86	<0.001	0.93	1.37	0.26	0.03	0.02	0.977	0.001
	Error Rates	71.99	<0.001	0.44	0.71	0.496	0.02	1.45	0.241	0.03

Note: Main effect of Cue/Flanker refers to the main effect of attentional cue for the alerting and orienting networks and the main effect of flanker type for the executive network.

## Data Availability

The data presented in this study are available on request from the corresponding author. The data are not publicly available due to privacy and ethical restrictions.

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
