# Peer review of "The Impact of Interpreting Training Experience on the Attentional Networks and Their Dynamics"

_brainsci, 2023, doi:10.3390/brainsci13091306_

Round 1

Reviewer 1 Report

Dear authors

Along with some great ideas discussed here, we would suggest you thinking of the readers who, after coming across with a digit in brackets, say [20], need to turn down the pages to see who it was, come back to read the article and again move to the References to see who was hidden behind [21]. [5] or any others. Why not to use (at least from time to time); as X [20] mentions/states/argues/says, .......? 

We sometimes see, e.g., [21,22] (comma, no space between), and sometimes - e.g., [26-27] (hyphenated). There should be uniformity.

The references are incorrect. E.g., the text suggests Morales [43], while in the References it is [41], Woumans is suggested [45], while it is [43], cowan should be [24-25] but not [26-27]. We cannot say how many errors are there in the References because, as said above, many of the sources are hidden behind the digits only.

Babcock, L.; Capizzi, M.; comes as [1] after [11] in the References, while mentioned the same team [12] in the text, line 90, referring Babcock as [45] and at the same time it is regarded as [46] in the manuscript.

Zhao, H.; Dong, Y. come as [2] after [15] in the References

why is [3] after [62]?

THE WHOLE REFERENCES MUST BE CHECKED AND CORRECTED. The references in the manuscript MUST be checked and corrected based on the References or vice versa.

There should be uniformity: either AoA or AOA (line 163)

lines 246, 259,274, 282, 295, 296, 310. we consider that there cannot be "A main effect" but can be THE main effect.

line 328. You need to think about an auxiliary verb: "suggesting that more incorrect responses made under center-cue conditions and incongruent conditions".

We would suggest adding a figure (axis, graph, table or alike) when speaking in subheading 3.2

lines 414-415. It is surprising that two different things are indexed the same way - P3: "early attentional processing (as indexed by larger P3 amplitude) and greater global interference control (as indexed by larger P3 amplitude).

line .  Burgoyne and his team members [90], mentioned in the text, cannot be found in the References because there are 87 sources. However, if you are patient and ready to search, you will find it under [87].

You need to work a bit to polish the language: punctuation and grammar.

Reviewer 2 Report

Review

“The impact of interpreting training experience on the

attentional networks and their dynamics”

General assessment:

·      The paper presents the results of an empirical study addressing the question of whether interpreting training contributes to the attentional networks of (late) bilinguals and the connection between interpreters’ attention and working memory. The investigation involved three groups of Chinese-English late bilinguals differing from each other in their more or less advanced level of interpreting training. The main results of the study results suggest: (i) that the alerting network is more efficient in the group with the longer interpreting training; (ii) that there is a negative correlation between the executive effect and the working-memory capacity in the trained group.

·      The topic of the investigation is clearly very relevant to the corresponding special issue. The study has been carried out by researchers with a high expertise in this field. The research question is presented in a very clear and precise way, as are the results. The discussion in Ch. 4 is transparent and clean. The paper is very well-written. The conclusions are stated in a clear and understandable manner.

For these reasons, I have no doubt about the publishability of the paper. However, there are some minor aspects that the authors may want to take a look at (see below) before submitting the final draft of the paper.

My overall recommendation is:

ACCEPT AFTER MINOR REVISION

Content:

- A general question that automatically comes to mind when considering on the one hand the relevant studies presented on p. 2-3 and the scope of the present study is whether the genetic (non-)relatedness between the languages investigated in one and the same study plays a role in producing results that tend to show the one or the other systematicity. The authors do not really make this explicit here, but the languages involved in the studies e.g. by Woumans et al. or Nour et al. are interrelated. Can the genetic relatedness between, say, French and Dutch (Woumans et al.) have an effect on the aspects considered here (vs., say, English and Chinese, especially considering the group with the least interpreting experience)? It would be nice to see a footnote or a short treatment of this at some point in the paper – maybe in 4.4, where some limitations of the present study are discussed.

- p. 10: “our findings showed the contribution of interpreting training experience” > I do not know if this is due to the language or to the content, but here you would rather say something like “showed an effect of interpreting training experience for …”. The training experience does not per se contribute to the dynamics: its effects or correlates are observable w.r.t. the variables addressed here.

Formal aspects:

- Abstract: The acronyms used to identify the three groups are used in the abstract without being introduced. Please use the full expressions here. These acronyms are presented on p. 3: While it is apparent that “IE” stands for “interpreting experience”, the authors should also provide the full description when introducing the three groups in the text.

- p. 1, “Interpreting is a highly demanding bilingual task in which interpreters continuously 29

switch between two highly activated” > The repetition of “highly” is infelicitous. Please replace one of these two occurrences of the word.

- p. 4: “Note: L1, first language; L2, second language, …” > Please use an equals sign (‘=’), not a comma, to make the acronyms explicit. This is very counterintuitive.

- 2.3.4, Title and text: Please use either “correlations analyses” or “correlation analyses” – but consistently – throughout the text.

Reviewer 3 Report

The impact of interpreting training experience on the attentional networks and their dynamics is an interesting manuscript investigating the relationship between groups of bilinguals with different levels of interpreting experience (more, less, or none).

While I have a background in second language research, my understanding of attention and working memory is not as well-grounded, still I hope that my questions and comments will be helpful especially since they pertain more to the statistics and description of the results. All comments can be found in the attached pdf file.

Best wishes

Dear authors,

I have commented on only minor issues with the English language, see minor comments in the pdf that is attached.

Round 2

Reviewer 1 Report

Thank you for your devotion and efforts in improving the quality of the manuscript. I also read the second reviewer's remarks and saw that you have done your best.

Author Response

We would like to express our sincere gratitude for the valuable comments you provided. Your comments have had a significantly positive impact on our research. We appreciate your patience and professionalism, and your suggestions have helped us to improve our manuscript. Once again, thank you very much for your contribution.

Reviewer 3 Report

Dear authors,

thank you for clarifying all your changes and for your explanations where I had misunderstood. I am impressed with how many changes that you have made to your manuscript and appreciate how these changes made the manuscript clearer for me to read.

All the best